# Unraveling the Mystery of Energy-Sensing Enzymes and Signaling Pathways in Tumorigenesis and Their Potential as Therapeutic Targets for Cancer

**DOI:** 10.3390/cells13171474

**Published:** 2024-09-02

**Authors:** Zeenat Mirza, Sajjad Karim

**Affiliations:** 1King Fahd Medical Research Center, King Abdulaziz University, Jeddah 21587, Saudi Arabia; 2Department of Medical Laboratory Sciences, Faculty of Applied Medical Sciences, King Abdulaziz University, Jeddah 21587, Saudi Arabia; skarim1@kau.edu.sa; 3Center of Excellence in Genomic Medicine Research, Faculty of Applied Medical Sciences, King Abdulaziz University, Jeddah 21587, Saudi Arabia

**Keywords:** cancer, energy-sensing enzymes, genes and signaling pathways, Warburg effect, anticancer drug

## Abstract

Cancer research has advanced tremendously with the identification of causative genes, proteins, and signaling pathways. Numerous antitumor drugs have been designed and screened for cancer therapeutics; however, designing target-specific drugs for malignant cells with minimal side effects is challenging. Recently, energy-sensing- and homeostasis-associated molecules and signaling pathways playing a role in proliferation, apoptosis, autophagy, and angiogenesis have received increasing attention. Energy-metabolism-based studies have shown the contribution of energetics to cancer development, where tumor cells show increased glycolytic activity and decreased oxidative phosphorylation (the Warburg effect) in order to obtain the required additional energy for rapid division. The role of energy homeostasis in the survival of normal as well as malignant cells is critical; therefore, fuel intake and expenditure must be balanced within acceptable limits. Thus, energy-sensing enzymes detecting the disruption of glycolysis, AMP, ATP, or GTP levels are promising anticancer therapeutic targets. Here, we review the common energy mediators and energy sensors and their metabolic properties, mechanisms, and associated signaling pathways involved in carcinogenesis, and explore the possibility of identifying drugs for inhibiting the energy metabolism of tumor cells. Furthermore, to corroborate our hypothesis, we performed meta-analysis based on transcriptomic profiling to search for energy-associated biomarkers and canonical pathways.

## 1. Introduction

Cancer, as a leading health problem globally, was diagnosed in an estimated 19.3 million new cases and accounted for approximately 10 million deaths in 2020; it is projected that it will be responsible for an estimated 13.1 million deaths in 2030 [1,2]. Cancer originates intracellularly due to genetic changes, mostly involving activated oncogenes or defective tumor suppressor genes, leading to the development of multiple cancer hallmarks, such as unrestrained growth and resistance to pro-death and antigrowth signals [3,4,5]. The conversion of normal cells into malignant cells and tumors progresses through multiple steps that have been categorized into six established hallmarks, including (i) proliferative signaling, (ii) escaping growth suppressors, (iii) resisting apoptosis, (iv) allowing replicative immortality, (v) triggering angiogenesis, and (vi) inducing invasion and metastasis [4]. In the last decade, evidence from studies has led to the inclusion of two further hallmarks of cancer progression, namely (vii) the reprogramming of energy metabolism and (viii) eluding immune destruction, thus extending the list to eight hallmarks [5]. Cancer metabolism is regulated by autonomous mechanisms within individual cells, along with metabolite availability in the tumor microenvironment (TME).

Energy-linked metabolic changes in cancer cells were first observed long before the discovery of oncogenes and tumor suppressors but have not been explored extensively. Energy homeostasis is crucial for cell fate, as metabolic regulation, cell proliferation, and apoptosis depend mainly on the equilibrium between anabolic and catabolic pathways. A recent study on tumor metabolism provided a further detailed view of the metabolic landscape of cancer cells, with deregulated energy metabolism becoming recognized as a cancer hallmark [5,6]. Neoplastic cells rewire their metabolism in response to external stimuli, enabling them to meet their increased demand for energy and macromolecule building blocks. Aerobic glycolysis, fatty acid oxidation, and anaplerotic reactions (replenishing critical biological reactions, e.g., tricarboxylic acid (TCA) cycle intermediates that have been extracted for biosynthesis) contribute substantially to biosynthetic anabolic pathways during cellular proliferation. Such ATP-consuming steps should be counterbalanced to maintain adenylate energy charge values within the physiological range, as their deregulation affects cell viability [7]. Therefore, in malignant cells, energy homeostasis is essentially preserved by means of continuous catabolism and mitochondrial ATP production [6].

For rapid proliferation, cancer cells require surplus energy fueled by atypical metabolic pathways. Most mammalian cells employ glucose as a fuel source, which is metabolized into pyruvate via a multistep glycolytic pathway. In healthy cells under regular aerobic conditions, most of this pyruvate enters the mitochondria and is oxidized via the Krebs cycle to produce the energy currency, ATP. However, in cancer cells, most of the pyruvate from glycolysis is directed away from the mitochondria to generate lactate via catalysis by lactate dehydrogenase (LDH), a process usually reserved for hypoxic states.

Contrary to mitochondrial glycolysis, lactate production in the presence of oxygen is called “aerobic glycolysis”, or the Warburg effect. Numerous signaling pathways involving PI3K/AKT signaling, LKB1/AMPK signaling, RTK signaling, fatty acid oxidation, glucose metabolism, glutamine metabolism, the folate cycle, branched chain amino acid metabolism, etc., further contribute to the Warburg effect and the metabolic phenotypes of cancer cells [8]. To cultivate a better understanding of glycolysis and other relevant reactions, many different aspects need to be researched in this complex process, which is influenced by multiple factors and variables. Biomarker discovery has been facilitated by breakthroughs related to advances in whole-exome and whole-genome sequencing along with customized panels of sequenced cancer-associated target genes, which facilitate genotyping, population-based genome-wide association studies, and the identification of single nucleotide variants [9]. To support the hypothesis of energy-sensors as potential tumorigenic biomarkers, transcriptomic meta-analysis was conducted followed by discussion of genes/proteins and canonical pathways associated with energy sensing and therapeutic possibilities targeting cancer cell energetics.

## 2. Transcriptomic Profiling and Pathway Analysis

We retrieved 2742 differentially expressed genes (DEGs) (722 upregulated and 2020 downregulated) from oncoDB (1135 breast cancer samples and 114 normal breast tissue samples) using a cut-off value of <1.00 × 10^−3^ for the adjusted *p*-value and log2-fold change (|log2FC|) > 1 (https://oncodb.org/cgi-bin/expression_norm_search.cgi (accessed on 10 February 2024)). Additionally, we downloaded 11 NCBI’s gene expression omnibus (GEO) datasets (GSE3744, GSE7904, GSE10780, GSE26910, GSE29431, GSE30010, GSE31138, GSE42568, GSE61304, GSE71053, and GSE111662), including 701 samples (356 breast tumor samples and 345 normal breast tissue samples), and derived 355 DEGs (77 upregulated and 278 downregulated) with a *p*-value < 0.05 and log2FC > 2. Ingenuity pathway analysis (IPA, QIAGEN Redwood City, CA 94063, USA) was conducted for breast cancer DEGs from the oncoDB and GEO datasets to predict pathways associated with energy, nutrients, and metabolites, including those associated with the tumor microenvironment, the integration of energy metabolism, AMPK signaling, RAR activation, HIF1α signaling, autophagy, sirtuin signaling, mitochondrial dysfunction, etc. (Table 1, Figure 1).

## 3. Shutting Down Mitochondrial Genes—Inhibition of Mitochondrial Biogenesis Pathways

Mitochondria convert calories from our diet into usable energy via oxidative phosphorylation, producing ROS as a lethal by-product, indicating that mitochondrial dysfunction plays a dominant role in multiple age-related disorders, including diabetes and cancer [10]. Mitochondria play an important role in apoptosis induced via ROS leakage from respiratory complexes, combined with the release of cytochrome c and diminished ATP synthesis [11]. Mitochondria can dynamically adjust their activity by decreasing oxidative phosphorylation and through changes in TCA cycle metabolism, mass, and biogenesis in order to meet cell physiological demands during tumorigenesis [12,13]. Mitochondrial biogenesis is regulated by eNOS, SIRTs, TORCs, and AMPK through the increased transcription of PGC-1α, NRFs, and mtDNA expression in order to support oxidative phosphorylation. However, mitochondrial biogenesis pathways were found to be inhibited in cancer cells, as energy is generated using aerobic glycolysis, not oxidative phosphorylation (Figure 2).

Additionally, numerous mutations in nuclear and mitochondrial DNA encoding metabolic enzymes have been reported in cancer [14,15]. The mitochondria function as metabolic hubs that integrate different metabolic routes in transformed cells [6]. Interestingly, the necessity for mitochondrial respiration varies with the stage of cancer, as shown by the recent study finding that primary and metastatic tumors exhibit divergent metabolic requirements [16]. Unlike nuclear DNA, the mitochondrial genome copy number (mtDNA-CN) varies within a cell and is significantly linked with the tissue’s bioenergetic requirements. POLGA (DNA polymerase γ A), a catalytic subunit of POLG, plays a vital role in mitochondrial dynamics and has been implicated in mtDNA-CN regulation. High methylation levels in exon 2 of POLGA have been linked with lower copy number in cells and, hence, with differentiation and cancer. Moreover, a decrease in mtDNA-CN has been linked to altered cellular morphology, decreased respiratory enzyme activity, and lower protein expression [17].

## 4. Genes/Proteins Associated with Energy Sensing

The regulation and maintenance of energy homeostasis is essential for normal physiology and cell survival and is linked to protein degradation and synthesis for the proper function of all cells [18]. Both protein synthesis (involving transcription and translation) and Ub-dependent protein degradation involve high energy expenditure [19,20]. However, autophagy and Ub-independent protein degradation are energy-saving processes [21,22]. Prokaryotes have a simpler streamlined genome, in which transcription and translation are coupled and occur simultaneously at the same place, offering several advantages, mainly assisting in gene regulation, while the energy required to drive transcription may be provided by the large-scale consumption of unstable nucleotide triphosphates during the translation process. Meanwhile, in eukaryotes, transcription and translation are spatially and temporally isolated, and splicing to remove introns takes place in the nuclear compartment prior to translation, while post-translation modifications occur in the cytoplasm [23,24]. Energy levels in and around cells must be carefully observed and attuned to ensure that energy intake and consumption remain within tolerable limits. The main fuel sources for cells are glucose/carbohydrates, followed by fats, proteins, and ketone bodies, which are converted into energy molecules like ATP, GTP, AMP, NAD^+^, etc., for cellular metabolic activities [25]. Several energy-sensing molecules have been demonstrated to detect variations in energy homeostasis and trigger the regulation of gene expression. Over-consumption of glucose is a central feature of cancer cells [26]. Cancer metabolism differs from that of normal cells, and a myriad of antiglycolytic drugs targeting various signaling pathways have been tested for their efficacy, and these molecules have demonstrated some success although along with some concerns. Recently, trends in developing novel anticancer drugs have shifted toward targeting molecules involved in cancer cell energetics.

Bioenergetics deals with energy flow in a biotic system and mainly involves the transformation of macronutrients—carbohydrates, fats, and proteins, which all contain chemical energy—into biologically functional energy forms [27]. Transformed cancer cells are constantly exposed to ROS, inducing random DNA mutations leading to genomic and chromosomal destabilization [28]. With the TCA cycle being the primary housekeeping pathway, any impairment induced by oncogenic mutations can trigger multiple metabolic rearrangements crucial for cell survival. Mutations in TCA genes (succinate dehydrogenase, isocitrate dehydrogenase, fumarate hydratase, etc.) have been linked with familial malignancies [29]. The AMPK, cytosolic guanylyl cyclase (cGC), phosphatidylinositol-5-phosphate 4-kinase-β, peroxisome proliferator-activated receptor, hypoxia-inducible factor 1, and Sirt1 proteins all respond to changing gene expression by sensing variations in the adenosine monophosphate (AMP), adenosine triphosphate (ATP), guanosine triphosphate (GTP), molecular oxygen, intracellular free fatty acid, and NAD^+^ concentrations, respectively [30,31] (Table 2). ATP and GTP are dynamic molecules that store and transfer energy for different metabolic processes in a cell. ATP is responsible for handling most of a cell’s energy needs, whereas GTP provides energy for protein synthesis, wherein the addition of one amino acid to a growing polypeptide chain consumes two GTP molecules. Energy metabolism pathways are influenced by enzymes/hormones that control physiology via the regulation of hunger, absorption, transportation, and the oxidation of food. Therefore, the integration of energy metabolism and cellular homeostasis is essential for the proper functioning of normal metabolism, and any changes in the concentrations of ATP and GTP can affect cellular homeostasis, which may lead to diseased conditions (Figure 3). For instance, rapidly proliferating cancer cells need a high concentration of GTP as fuel, which is thus compromised in its primary tasks of protein synthesis and acting as a signaling molecule in normal cells [31,32].

### 4.1. Kinases, Enzymes, and Sensors of Energy Metabolism

Biochemically, kinases are enzymes that catalyze phosphate group (PO_4_^3−^) transfer from high-energy, phosphate-donating molecules (usually ATP molecules) to specific substrates through phosphorylation. Kinases are activated by increases in the concentration of AMP as well as ADP. Adenosine kinase (ADK), an evolutionarily conserved phosphotransferase, converts the purine ribonucleoside adenosine into 5′-adenosine-monophosphate (AMP). ADK continuously monitors ATP, ADP, AMP, and other adenyl phosphate nucleotide levels inside the cell, thereby regulating the amount of energy expended at the cellular level and, hence, playing an important role in cellular energy homeostasis [33,34].

Kinase inhibition is an important approach for inhibiting upregulated enzymes in the treatment of complex diseases like cancer. Abelson tyrosine kinase (ATK), AMP-activated protein kinase (AMPK), adenylate kinase (AKE), and PAS kinase (PASK) are the most common cancer-associated kinases. ATK is overexpressed in chronic myelogenous leukemia (CML). Imatinib (Gleevec) can bind to the catalytic site of ATK, thereby blocking the enzyme’s ability to phosphorylate targets, thus being useful in the initial treatment of CML [35]. AMPK is a phylogenetically conserved sensor of nutrient and energy status and sustains cellular energy homeostasis in addition to numerous other physiological functions, like the regulation of mitochondrial biogenesis and clearance, cell polarity, autophagy, and cell proliferation [36]. Both viruses and tumor cells have established means to downregulate AMPK, permitting them to evade its growth-limiting restraints, with some evidence indicating that AMPK can curb various RNA viruses by restricting fatty acids [36,37,38]. AMPK is thus one of the most promising emerging preventive and therapeutic targets for cancer and other major chronic problems, including obesity, diabetes, and inflammation [39]. AKE can swiftly replenish ATP, and this reaction (2ADP + adenylate kinase → ATP + AMP) is significant, as AMP, a product of the adenylate kinase (myokinase) reaction, is a strong stimulator of glycolysis. We also further discuss AMPK signaling later. PASK is a nutrient and bioenergetic sensor that responds to glucose availability and controls glucose homeostasis [40].

### 4.2. Sirtuin, a Metabolic Sensor

Sirtuin 1 (SIRT1), a metabolic sensor, is an NAD^+^-dependent protein deacetylase which regulates chromatin structure and gene expression to modulate glucose metabolism (gluconeogenesis, glycolysis, and insulin sensitivity), lipid energy metabolism (fatty acid oxidation and cholesterol metabolism), stress response, and tumorigenesis [41,42]. SIRT1 enables excess energy to be stored in the liver by synthesizing glycogen and forming lipid droplets. However, during fasting and exercise, the energy level goes down and cellular levels of NAD^+^ increase, stimulating SIRT1 activity [43], while reduced SIRT1 activity was observed with a high-fat diet, leading to a high-energy status that diminishes cellular NAD^+^ levels [44]. The balance of energy within cells is critical for their survival and homeostasis. The most energy-intensive process for eukaryotes is ribosome biosynthesis, which responds to changes in energy status via a complex machinery. Energy-dependent nucleolar silencing complex (eNoSC) is a protein complex which senses energy levels and regulates rRNA transcription. It can restore energy levels under low-glucose conditions. eNoSC contains SUV39H1 and SIRT1, which are essential for energy-dependent transcriptional repression, indicating that any variation in the NAD^+^/NADH ratio would induce deacetylation and demethylation and subsequent chromatin silencing. It facilitates cellular energy balance restoration by restricting rRNA transcription, thereby shielding cells from energy deprivation-supported apoptosis [20] (Figure 4).

### 4.3. Soluble Guanylyl Cyclase, an ATP Sensor

Soluble guanylyl cyclase (sGC), a receptor for nitric oxide (NO), is an ATP sensor whose sensitivity to NO is controlled by intracellular ATP (ATPi) levels. Under physiological conditions, ATP binds to an allosteric site in sGC and inhibits NO/sGC signaling, while under pathological conditions with reduced ATPi, sGC freely binds to NO, which leads to the regulation of ATP supply and demand via NO/sGC signaling. NO synthesis modulation is linked to cardiovascular disorders and hypertension [45,46]. The regulation of cGMP synthesis by sGC is controlled by GTP, ATP, NO, and allosteric activators like YC-1, a novel activator of platelet guanylate cyclase [47]. However, the mechanisms linking NO/sGC and energetic signaling remain undefined.

### 4.4. Phosphatidylinositol-5-phosphate 4-kinase-β, a GTP Sensor

Phosphatidylinositol-5-phosphate 4-kinase-β (PI5P4Kβ), a lipid kinase, is responsible for detecting the GTP supply, an energy source which fuels the uncontrolled proliferation of tumor cells [31]. Compelling evidence links PI5P4Ks (α and β) to cancers where p53 deficiency is part of their pathology [48,49]. For their division, cancer cells need to know the availability of fuel; thus, by interfering with PI5P4Kβ’s ability to sense intracellular GTP availability, one can slow down the proliferation of these cells.

### 4.5. Glucokinase, a Glucose Sensor

Type 2 diabetes and other human disorders including cancer are primarily caused by dysregulations in the glucose homeostasis processes, which include glucose sensing, importation, storage, and mobilization systems [50]. Intracellular glucose sensing takes place in pancreatic and hepatic cells, driven by glucokinase (GCK). GCK has a poor affinity for glucose and diverts glucose-6-phosphate into glycogen synthesis or glycolysis only in glucose-abundant situations. GCK plays a central role in maintaining organismal glucose homeostasis. Due to having a similarly poor affinity, the GLUT2 transporter imports glucose actively only during a high glycemic state. Owing to its bidirectional features, it can also transfer glucose from hepatocytes into the blood flow under hypoglycemic conditions when intrahepatic glucose levels are elevated by hepatic gluconeogenesis and glycogen breakdown [51].

### 4.6. Glutamine Metabolism

Glutamine maintains mitochondrial metabolism and serves as a carbon source for lipid and metabolite synthesis via the TCA cycle, as well as a source of nitrogen for the synthesis of nonessential amino acids and nucleotides [52,53,54]. Glutaminolysis is the breakdown of glutamine via the activity of glutaminase glutamine dehydrogenase and a group of transaminases into downstream metabolites like glutamate and α-ketoglutarate, which are much-needed intermediates fueling the TCA cycle in tumors. Like glycolysis, glutaminolysis supplies cancer cells with both ATP and key precursors for continuous biosynthesis and accelerated proliferation [55]. Cancer cells often use glutamine as an alternative fuel source to meet their higher energy demands. Glutamine enters the mitochondria and is used to restore intermediates in the Krebs cycle or to generate more pyruvate through malic enzymes, or else to generate building blocks (amino acids and nucleotides) required for increased cell growth. During rapid proliferation, glutamine metabolism in the TME acts against oxidative stress in maintaining redox homeostasis by producing glutathione, an antioxidant that prevents reactive oxygen species (ROS) accumulation. Malignant cells may develop a strong dependence on glutamine, further encouraging proliferation [56,57]. It has been reported that under low glucose levels, glioblastoma cells start utilizing glutamine. If glutamine metabolism is disturbed, the cells cannot function without glucose. Therefore, in the absence of glucose, cells use amino acids to try to maintain homeostatic and energy balance [26]. High glutamine consumption has long been implicated in cancer cell growth; hence, glutamine metabolism in cancer is becoming an attractive research subject.

## 5. Pathways Associated with Energy Sensors and Metabolism

Metabolic processes include energy generation, energy consumption, and energy sensing, and homeostasis is a complex process involving switching between multiple pathways depending on cellular requirements. In general, glycolysis (ATP from glucose), the mitochondrial respiration (the TCA cycle and oxidative phosphorylation, along with ATP from pyruvate) system, and the phosphagen (quick ATP from creatine phosphate) system are used to generate active energy. The regulatory functions of these enzymes and intermediates in controlling food consumption and energy equilibrium are critical to understanding tumor biology [13,30]. In addition to basic glycolysis and mitochondrial respiration, a number of signaling and metabolic pathways have been reported to regulate energy metabolism, including the AMPK, ERK/MAPK, HIF1α, glutamine, glutaminergic receptor, p53, PI3K/AKT, ULK1 (autophagy), and mTOR signaling pathways [6].

### 5.1. AMPK Signaling Pathway

In order to appropriately maintain energy homeostasis, AMPK activates ATP-producing catabolic pathways while deactivating ATP-consuming anabolic pathways [36]. The functional coupling of electron transfer to ATP synthesis (oxidative phosphorylation) is vital throughout explicit tumor progression phases when tumor cells face oxygen and nutrient scarcity. Consequently, ATP exhaustion can activate the fuel-sensing enzyme AMPK, supporting metabolic hypoxic adaptation, thereby provoking autophagy and mitochondrial biogenesis. Nevertheless, functionally active AMPK may represent an inhibiting factor during the initial phases of tumorigenesis because it suppresses cell proliferation, inhibiting the mTOR pathway (Figure 5). The precise molecular steps by which AMPK affects carcinogenesis are contentious. The function of different AMPK isoforms, the circumstances in which AMPK may influence tumor progression, and how different tumor suppressors or oncogenes control this kinase are subjects which require in-depth explanation. AMPK behaves as a negative regulator of aerobic glycolysis (the Warburg effect) and suppresses tumor growth [58].

AMPK is considered an *oncojanus,* being a double-edged sensor of cancer cell metabolism and involved in the regulation of tumorigenic potential, since it exhibits opposing effects in different cell contexts and throughout definite tumor progression phases. As a pro-tumorigenic inducer, active AMPK may promote autophagy and metabolic flexibility, ultimately leading to cell survival. However, AMPK may have an antitumorigenic effect by causing cell cycle arrest and inhibiting mTORC1 signaling [6]. Developing a better understanding of AMPK’s therapeutic window is imperative to combatting tumor development.

Most compounds currently under investigation indirectly control the AMP/ATP ratio via the inhibition of mitochondrial oxidative phosphorylation. AMPK activation mediated by the extensively used antidiabetic drug metformin exhibiting a reduced risk hazard of cancer in diabetic patients has been reported [59]. Metformin inhibits respiratory complex I without activating ROS production. Hence, it can be repurposed for cancer treatment; however, whether its antitumorigenic potential arises from its action at the cellular or systemic level, or in a synergistic way, is still disputed [60]. By phosphorylating the essential RAF/KSR family kinases, AMPK signaling can also reversibly regulate overactive MAPK signaling in cancer cells. This allows for regulating not only carcinogenesis, but also the effects of targeted cancer therapies that target MAPK signaling.

### 5.2. ERK/MAPK Signaling Pathway

The ERK (extracellular-regulated kinase)/MAPK (mitogen-activated protein kinase) pathway mainly regulates growth and differentiation (Figure 6). It starts with the binding of ligands to membrane-bound receptor tyrosine kinases (RTKs) to form a signaling complex with growth factor receptor bound protein 2 (GRB2), son of sevenless (SOS), and Shc, with subsequent activation of RAS, initiating a MAPK kinase cascade. In the cytoplasm, ERK can phosphorylate and regulate cytoskeleton proteins, ion channels/receptors, and translation, while in the nucleus, it regulates gene transcription through interaction with ELK-1, STAT1, ETS, MYC, CREB, c-Fos, and SRF molecules. Malignant cells reprogram their glucose metabolism to aerobic glycolysis through altered signaling pathways including MAPK/ERK to promote proliferation and invasion [61]. The Warburg effect is promoted by the MAPK/ERK pathway via the activation of RAS oncoproteins (HRAS, KRAS, and NRAS) [62]. The MAPK/ERK pathway, a key regulator of the Warburg effect, could be a potential drug target for inhibiting tumorigenesis and facilitating normal cellular functions [63].

### 5.3. HIF-1α Signaling Pathway

HIF-1 is a basic helix–loop–helix transcription factor which activates genes to induce homeostatic responses to hypoxia, a low-oxygen condition that occurs in cancer, inflammation, and diabetes as well as at high altitude [64]. Oxygen is fundamental for aerobic mitochondrial respiration and bioenergetics for cellular functions. However, hypoxia is characteristic of the tumor microenvironment, where hypoxia-inducible factor 1 alpha (HIF-1α) is required to control metabolic reprogramming via close crosstalk between the mitochondria and HIF-1α [65,66]. When oxygen is present, HIF1 binds to E3 ubiquitin ligase and von Hippel–Lindau (VHL) tumor suppressor protein. The degradation of HIF1α is activated by a complex of VHL, Cul2, RBX1, elongin-B, elongin-C, ubiquitin-conjugating enzyme, and ubiquitin-activating enzyme. Under hypoxic conditions, HIF1-α escapes ubiquitylation and degradation, as this subunit is not recognized by VHL; therefore, the accumulation and dimerization of HIF1-β initiates the transcription of target genes in the nucleolus. HIF1-α activates VEGF, GLUT1, LDHA, Epo, and NOS, promoting angiogenesis, glucose transport, the glycolytic pathway, erythropoiesis, and vasodilation, respectively [67].

### 5.4. Glutamine and Glutaminergic Receptor Signaling Pathway

While glutamine is necessary as an alternate fuel for ATP production via TCA, it also plays a critical role in a number of other pathways, including those that produce amino acids (especially proline and asparagine), nucleotides (de novo synthesis of purine and pyrimidine), and fatty acids along with those that activate mTOR and hexosamine biosynthesis [56]. The metabolism of glutamine in the TME may influence the growth and development of cancers [57]. Interestingly, glutamine metabolism in TME is associated with epigenetic alterations, as tumor areas with low concentrations of glutamine experience more histone hypermethylation owing to the reduction in α-ketoglutarate levels, resulting in significant BRAF inhibitor resistance in V600E*BRAF* melanoma cells. Even though glutamine is required for several tumor types, whether glutamine deprivation induces increased apoptosis or suppressed cell proliferation varies between cell types, and this has implications for glutamine-metabolism-related therapeutic approaches [68].

### 5.5. p53 Signaling Pathway

p53 is a tumor suppressor protein and a key transcriptional regulator which is activated by DNA damage, UV radiation, and hypoxia under stress conditions, including cancer. Basically, p53-dependent pathways remove damaged cells, either via apoptosis or cell-cycle checkpoint arrest. However, inhibition of the p53 signaling pathway or loss of p53 function contribute to DNA repair, cell cycle progression, and angiogenesis [69]. p53 regulates energy metabolism, inhibits glycolysis, and promotes oxidative phosphorylation by activating AMPK, PTEN, tuberous sclerosis complex 2 (TSC2), and the TP53-induced glycolysis and apoptosis regulator (TIGAR). This p53-induced activation of AMPK can also promote the phosphorylation and acetylation of p53, forming a positive feedback regulatory loop. Concurrently, p53 also limits the uptake of energy molecules by cancer cells, greatly impeding tumor growth [70]. Wild-type p53 has a tumor-suppressive biological response, in contrast to mutant p53, whose response leads to metastasis and therapy resistance. p53 is mutated in most human cancers. The loss of p53 pathway function provides malignant cells with a survival advantage to evade the continual of oncogenic signals and DNA damage to endure abnormal proliferation.

TP53 mutations can result in a stable protein with different pro-tumorigenic outcomes like a gain of oncogenic function, resistance to therapy, and poor prognosis. Accumulation of p53 mutants in tumors is ascribed to their ability to evade degradation by the proteasome [71]. The increased stability of mutant p53 primarily results from the inability of mutant p53 to transcriptionally control the target genes of wild-type p53, consequently revoking the negative feedback loop via Mdmd2. An alternative mechanism that controls mutant p53 protein levels is autophagy, the intracellular clearance process that acts upon damaged cellular components and recycles components to compensate for energy expense and establish cellular homeostasis. Metabolic stress such as glucose restriction can result in mutant p53 degradation through the autophagy machinery, and this seems to be dependent on p53 being deacetylated [72].

### 5.6. Autophagy Pathway

Autophagy or macroautophagy is a basic catabolic mechanism whereby damaged or dysfunctional cellular components are engulfed and degraded in lysosomes [73]. It begins with the birth of the phagophore and ends with the death of the autophagosome [74]. It is an intracellular recycling process in which a basal level of metabolites is maintained under adverse conditions. Nutrient deprivation during cancer and other pathological processes such as stress and infection activate autophagy through the negative regulation of the ULK1 complex at the endoplasmic reticulum and mTOR (AMPK and p53 signaling). The ULK1 complex, along with the PI3K complex and multiple ATG proteins, regulates autophagosome formation. LAMP proteins regulate the maturation of autophagosomes. The specific interaction of the HOPS (homotypic fusion and protein sorting) complex with Q-SNARE STX17 promotes autophagosome–lysosome fusion [75].

### 5.7. PI3K/AKT/mTOR Signaling Pathway

Phosphatidylinositol 3-kinase (PI3K), a member of the lipid kinases, regulates cytokines, growth factors, and extracellular matrix proteins [76]. PI3K interacts with receptor tyrosine kinases, the adapter molecules GAB1-GRB2, and the JAK kinase to activate 3- phosphoinositide-dependent kinase 1. This activated PDK1 phosphorylates AKT, which either phosphorylates the BAD/Bcl-XL complex to inhibit apoptosis and favor cell survival or activates IkappaB kinase to help cell survival [77,78]. PI3K/AKT is involved in energy storage, cell survival, cell growth, protein synthesis, biogenesis, vasodilation, and angiogenesis [79,80,81]. mTOR, a serine/threonine kinase, is the main target of PI3K/AKT and activated mTOR complex 1 signals for cell cycle progression and energy storage [82]. However, during tumorigenesis, mTORC1 is inhibited and autophagy is induced for nutrient and energy generation to promote the survival of cells under adverse conditions [83] (Figure 7).

The mammalian or mechanistic target of rapamycin (mTOR) kinase, as part of mTOR complex 1 (mTORC1), regulates cellular energetics by triggering several anabolic processes comprising lipid and protein synthesis [84]. The physiological functions of the mTOR complex vary according to the tissue in which it is being expressed. It regulates metabolic and energy homeostasis associated with the gut–brain axis. In response to peripheral inputs pertaining to energy status, mTOR modulates various hypothalamic energy balance regulation circuits [85]. The mTOR pathway is controlled at the level of the hypothalamus by the nutritional status, which responds to nutrient availability and the hormonal environment that governs the central nervous system’s neuronal circuits, which in turn control metabolism and energy balance. mTOR signaling is influenced by body weight and food intake [86]. Interestingly, hypothalamic interplay exists for the regulation of energy balance between the AMPK and mTOR pathways.

## 6. Therapeutic Routes Involving Targeting of Energy Sensing

### 6.1. Metabolic Therapy: The Inhibition of Tumor Cell Energy Metabolism

Rapidly proliferating malignant cells need large amounts of energy through fast tumor cell energy metabolism; therefore, various energy metabolism pathways comprising glucose transport, TCA, glutaminolysis, mitochondrial respiratory chain oxidative phosphorylation, and the pentose phosphate pathway (PPP) are substantially altered in cancer cells. The well-established “Warburg effect” in cancer cells has been shown to have functional characteristics of increased glucose intake, a high glycolysis rate, and a preference for aerobic glycolysis (converting most of the glucose to lactic acid) for energy, even in the presence of oxygen [87,88,89]. For fast growth and division, a tumor cell must double its genome, proteins, and lipids, so cells need to convert extracellular nutrients like glucose and glutamine into biosynthetic precursors to facilitate enhanced macromolecular biosynthesis activity during tumorigenesis [90,91,92] (Figure 1). The broader context of the Warburg effect is that “metabolic transformation” is required for tumorigenesis [90], and thus, targeting energy metabolism for cancer treatment (metabolic therapy) might be a very effective approach in this regard [88].

In addition to glycolysis, the Warburg effect plays a significant role in other metabolic pathways, like the PPP and associated ribose production pathways for nucleotide biosynthesis, serine synthesis, hexosamine biosynthesis, as well as lipid synthesis via the TCA cycle. The Warburg effect is a phenomenon associated with tumors that can be therapeutically targeted. The recent literature suggests that FDA-approved histone deacetylase inhibitors (panobinostat, vorinostat, and romidepsin) elicit metabolic reprogramming and may strongly disrupt super-enhancers linked with the Warburg effect [93]. However, regulation of the Warburg effect at the epigenetic level is not yet well understood.

Recent tumor metabolism studies presented innovative techniques for metabolic control and deciphering the role of key enzymes in tumor metabolism, leading to new diagnostic and therapeutic approaches. For example, Kherlopian et al. used the scanning technique of fluoro-deoxy-glucose-based positron emission tomography to test the landscapes of tumor aerobic glycolysis as diagnostic markers and determine the appropriate treatment [94].

Hexokinase, phosphofructokinase, and pyruvate kinase are the chief regulatory stages for glycolysis flux cadence, as they catalyze the three highly spontaneous irreversible reactions in glycolysis [95]. Glycolysis is regulated in part by the concentrations and turnover rates of these enzymes. Pyruvate dehydrogenase kinase converts pyruvate into lactate, a substrate for aerobic glycolysis.

Metabolic therapy is a promising strategy, as it is more specific than oncogene-targeted approaches because cancer cells are more sensitive to metabolic inhibitors than healthy cells [88,96] and are comparatively free of explicit signaling or epigenetic dysfunctions. However, the dual-hit approach of combining energy metabolism inhibitors with other anticancer inhibitors might be a more effective cancer therapeutic strategy than using energy metabolism inhibitors alone [97]. In vitro studies have shown delayed growth and cell death upon inhibiting glycolysis and/or oxidative phosphorylation [96].

The common functional characteristics of tumor cells include the fact that they devour a large quantity of glucose, sustain a high glycolysis rate, and convert most of the glucose into lactic acid, even in aerobic conditions, compared with healthy cells (the Warburg effect). Moreover, cancer cells display significant modifications in numerous energy metabolism pathways, including glucose transport, the TCA cycle, mitochondrial respiratory chain oxidative phosphorylation glutaminolysis, and the PPP. We primarily reviewed the existing knowledge about how, in cancer, various oncogenes including c-Myc, HIF-1, and p53 dysregulate the proteins/enzymes involved in the crucial regulatory steps of glucose transport, glycolysis, the TCA cycle, and glutaminolysis. Macrophage modulation through transcriptional rewiring using phytocompounds that affect glycolysis and OXPHOS of M1 macrophages to boost IL-1b production has been shown to be effective for macrophage-associated inflammatory diseases like cancer [98]. Oncogene-induced dysregulation of the glucose transport and energy metabolism pathways, along with the loss of tumor suppressor activity, has been identified as a critical indicator for cancer diagnosis and as a novel drug target for the development of anticancer therapeutics.

The gut microbiota might play significant roles in cancer development and the tumor microenvironment. Studying the gut microbial metagenome and intratumor microbiome has opened up new avenues to understand the interactions among diet, probiotics, cancer, and immunotherapeutic response and represents an emerging research paradigm for future exploration relating to anticancer immunotherapies [99,100].

### 6.2. New Generation of Selective JAK Inhibitors

The JAK-STAT pathway functions as a hub for many different signaling networks, exhibiting potential for crosstalk with other pathways. JAK-STAT signaling is constitutively activated, plays a pivotal role in myeloproliferative and inflammatory syndromes, and suggests an additional rationale for disease management via the repurposing of clinically available AMPK activators. AMPK activation can restrict JAK-STAT-dependent signaling via numerous pathways [101].

### 6.3. Nucleic Acid Sensing

Nucleic acid sensing is a crucial mechanism in gene and immunotherapies that target cancer through therapeutic nucleic acids or genetically engineered immune cells. Nucleic acid sensing exerts both pro- and antitumor effects at different stages of tumorigenesis, supports immune cells in priming desirable immune responses during therapies, and impacts the competence of gene therapy by inhibiting translation [102]. Cyclic guanosine monophosphate-adenosine monophosphate synthase (cGAS) senses accumulated DNA in cells and becomes activated upon binding to dsDNA. In an energy-consuming process, cGAS transforms ATP and GTP into cyclic GMP–AMP (cGAMP), a secondary messenger. cGAMP, along with other cyclic dinucleotides, conveys the signal downstream to the endoplasmic reticulum protein named stimulator of interferon genes (STING). Then, the signal cascades and finishes in interferon regulatory factor 3 (IRF3) and NF-κB targets in the nucleus, leading to the secretion of type-I interferon, which is vital to tumor-specific T cells [103,104]. Similarly, the RIG-I-like receptor (RLR)-MAVS pathways, responsible for cytosolic RNA sensing, have been targeted for cancer treatment in preclinical translational research [105]. Comprehending the interplay of distinct nucleic acid sensing mechanisms can facilitate the creation of novel strategies for the optimization of cancer immunotherapy.

## 7. Conclusions

Cancer research has received increasing attention in the last couple of decades, and several abnormalities in genes, proteins, and signaling pathways have been identified. Many antitumor drugs have been designed and screened for cancer therapy. However, it remains a major challenge to design drugs that specifically target malignant cells. Recent attention has been directed toward the contribution of energetics to cancer development, and energy-sensing enzymes have emerged as possible targets for cancer therapy. Studies based on energy metabolism have found increased glycolytic activity and decreased oxidative phosphorylation in cancer cells (the Warburg effect). The signaling pathways regulated by energy molecules and their sensors communicate with the extracellular environment via downstream pathways and exert effects on proliferation, apoptosis, autophagy, angiogenesis, and metabolic stress resistance, the processes most altered during carcinogenesis. The role of energy homeostasis in the survival of normal as well as malignant cells is critical; therefore, fuel intake and expenditure need to be balanced within acceptable limits. Thus, energy-sensing enzymes that detect disruptions in glycolysis, AMP, ATP, or GTP levels might be promising candidates for designing energy metabolism-inhibiting drugs to be used as specific anticancer therapies. We have comprehensively reviewed the common energy mediators and energy sensors along with their metabolic properties and mechanisms related to the carcinogenesis process. We also explored the possibility of using energy-sensing enzymes as potential targets in designing drugs for the inhibition of tumor cell energy metabolism.

## Figures and Tables

**Figure 1 cells-13-01474-f001:**
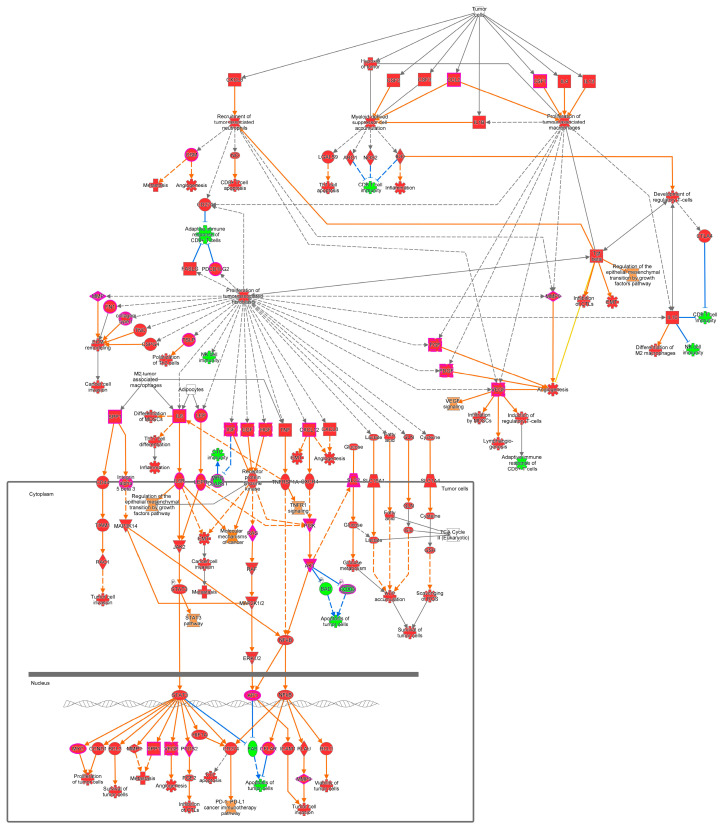
The tumor microenvironment pathway is predicted to be altered (z-score = −2.77) and closely associated [−log(*p*-value) = 8.83] with breast cancer. Description: AKT activates NFκB and SLC2 and inhibits BAD and FOXO via phosphorylation, where BAD and FOXO induce the apoptosis of tumor cells. ARG1 contributes to CD8 T-cell immunity; BCL2 promotes the survival of tumor cells; CCL2, CSF1, CSF2, and CSF3 enhance myeloid-derived suppressor cell accumulation and the proliferation of tumor-associated macrophages; CCND1 stimulates the proliferation of tumor cells; CD274 is responsible for the adaptive immune response of CD8+ T cells; CTL promotes apoptosis; and PD-1 is involved in the PD-L1 cancer immunotherapy pathway. CXCL12, CFCL8, FGF, MMP9, OSM, and VEGF are causative for angiogenesis. STAT3 regulates the expression of BCL2, CCND1, CD274, FAS, HIF1A, MMP2, MYC, PTGS2, SPP1, and VEGF, whereas NFκB regulates the expression of AP1, BCL2, CD274, CFLAR, and PLAU, which are responsible for tumor cell proliferation, viability, angiogenesis, and metastasis.

**Figure 2 cells-13-01474-f002:**
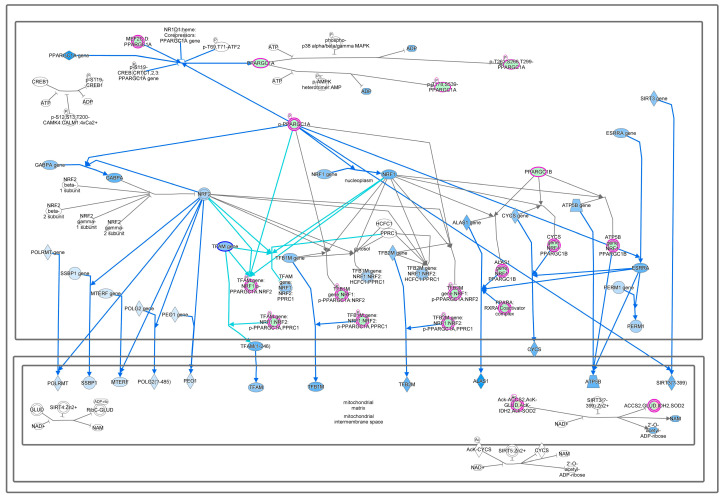
Mitochondrial biogenesis pathways. Downregulation of differentially expressed genes ACSS2, CHD9, IDH2, SOD2, MEF2C, PPARA, PPARGC1A, and PPARGC1B causes inhibition of the mitochondrial biogenesis process in breast cancer.

**Figure 3 cells-13-01474-f003:**
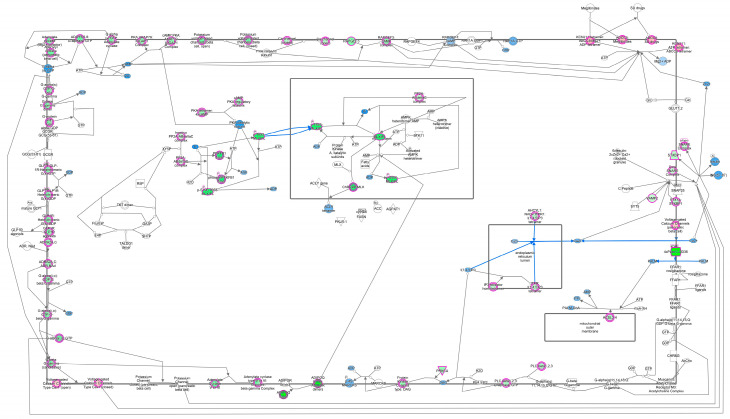
Integration of energy metabolism includes multiple events and pathways such as glucagon signaling in metabolic pathways, PKA-mediated phosphorylation, insulin stimulating increased expression of xylulose-5-phosphate (Xy-5-P), the AMP kinase (AMPK)-mediated response to elevated AMP, dephosphorylation of key metabolic factors by PP2A, and the transcriptional activation of metabolic genes by ChREBP.

**Figure 4 cells-13-01474-f004:**
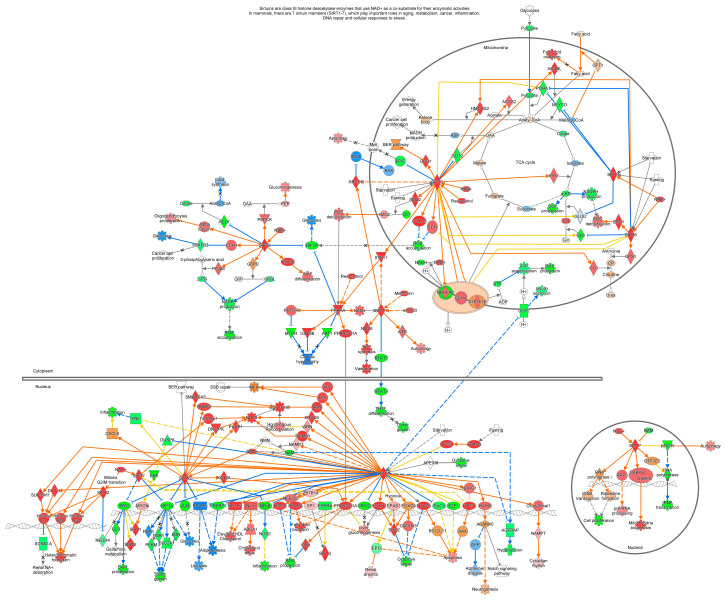
Sirtuin signaling pathway. Description: NAD^+^ activates SIRT1–SIRT7. SIRT1 converts NAD^+^ to NAM. SIRT1 activates ATG, Clock-Bmal1, DOT1L, EPAS1, FOXO1, FOXO3, FOXO4, HSF1, Histone H1, Histone H3, Histone H4, NAMPT, NBN, NDRG1, NFE2L2, NOS3, NR1H4, Nr1h, PPARA, PPARGC1A, RARB, STK11, SUV39H1, TRIM28, WRN, XPA, XPC, XRCC6, and ZBTB14 and inhibits CRTC2, DUSP6, E2F1, FOXO3, HIF1A, NFκB, PPARG, SREBF1, STAT3, TP53, Trp73, and UCP2. Solid and dashed lines indicate direct and indirect relationship, respectively. The predicted relationships are denoted by different colors, wherein orange means “leads to activation”; blue indicates “leads to inhibition”; yellow implies that findings are inconsistent with the state of downstream molecules, and grey depicts that the effect is not predicted.

**Figure 5 cells-13-01474-f005:**
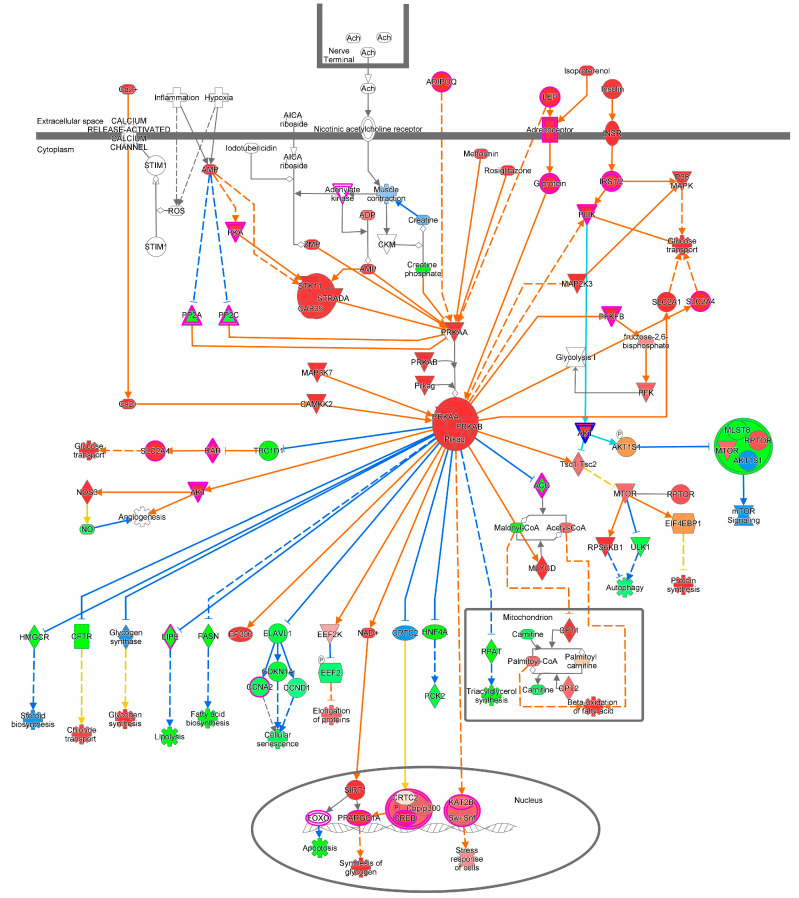
AMPK signaling pathway. AMPK activates AKT, MAP2K3, MLYCD, NAD^+^, PCAF-SWI/SNF, PFKFB, PI3K, SLC2A1, SLC2A4, and Tsc1-Tsc2 and inhibits ACC, CFTR, CRTC2, ELAVL1, FASN, glycogen synthase, HMGCR, HNF4A, LIPE, PPAT, and TBC1D1 via the phosphorylation of ACC, AKT, EEF2K, EP300, glycogen synthase, MLYCD, PFKFB, and PPAT. MTOR activates EIF4EBP1 and RPS6KB1 and inhibits ULK1 via phosphorylation. ULK1 induces autophagy. EEF2 is activated upon phosphorylation by EEF2K, which induces protein elongation. NOS3 is activated following phosphorylation by AKT1, promoting angiogenesis.

**Figure 6 cells-13-01474-f006:**
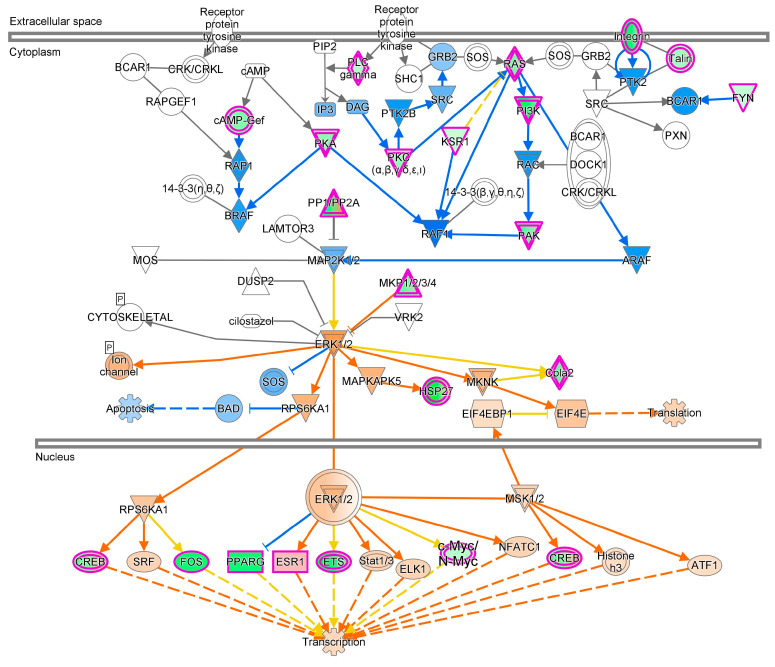
ERK/MAPK signaling pathway. Description: MAP2K1/2 activates ERK1/2 via phosphorylation. ERK1/2 activates Cpla2, ion channels, MAPKAPK5, and MKNK and inhibits SOS via phosphorylation, whereas the ERK1/2 dimer activates ELK1, ESR1, ETS, MSK1/2, STAT1/3, and c-Myc/N-Myc and inhibits PPARG via phosphorylation. Transcription is induced by ATF1, c-Myc/N-Myc, CREB, ELK1, ESR1, ETS, FOS, Histone H3, NFATC1, PPARG, SRF, Stat1/3, and EIF4E.

**Figure 7 cells-13-01474-f007:**
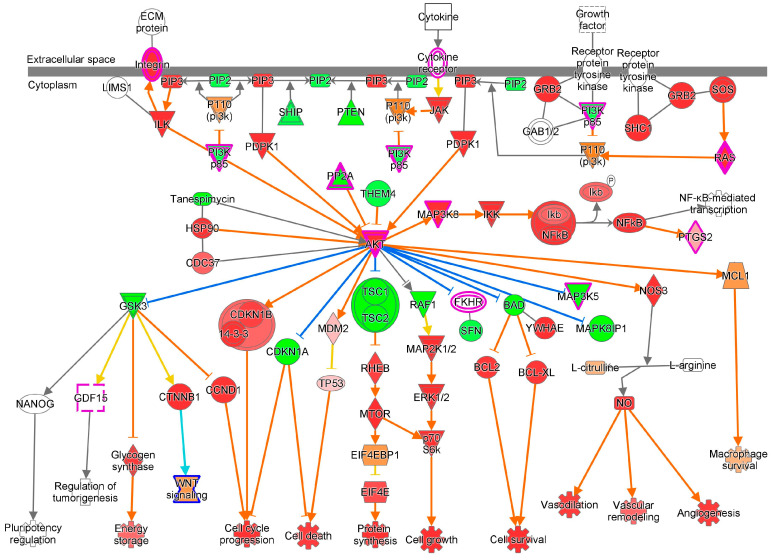
PI3K/AKT signaling pathway. AKT activates nitric oxide through NOS3 and promotes vasodilation and angiogenesis. AKT interacts with CDKN1B, MAP3K8, MAPK8IP1, MDM2, and RAF1 proteins to form complexes and inhibits BAD, FKHR, GSK3, MAP3K5, MAPK8IP1, RAF1, and TSC1/2 to promote cell survival and growth, cell cycle progression, energy storage, protein synthesis, vasodilation, and angiogenesis.

**Table 1 cells-13-01474-t001:** Energy-, nutrient-, and metabolite-associated canonical pathways in breast cancer and their component molecules as predicted by the ingenuity pathway analysis tool.

Ingenuity Canonical Pathways	−log (*p*-Value)	z-Score	Molecules
Tumor Microenvironment Pathway	8.83	−2.77	AKT3, CCL2, CFLAR, COL1A1, COL1A2, COL3A1, CSF1, CXCL12, EGF, FGF1, FGF2, FGF7, FN1, FOS, FOXO1, FOXO4, HGF, HLA-E, IGF1, IL6, IL6R, ITGB3, JUN, LEP, LEPR, MMP1, MMP10, MMP11, MMP13, MMP24, MMP28, MMP3, MMP9, MRAS, MYC, OSM, PDCD1LG2, PDGFA, PDGFD, PIK3C2G, PIK3R1, PTGS2, RASD1, RRAS2, SLC2A4, SPP1, TSLP, VEGFD
ERK/MAPK Signaling	4.68	−2.95	CREB3L1, CREB3L4, CREB5, DUSP1, DUSP6, ELF5, ESR1, ETS1, ETS2, FOS, FYN, HSPB1, HSPB7, ITGA1, ITGA10, ITGA6, ITGA7, ITGA9, ITGB3, ITGB4, ITGB8, KSR1, MRAS, MYC, PAK3, PAK5, PIK3C2G, PIK3R1, PLA2G4A, PLA2G5, PLCG2, PPARG, PPM1J, PPM1L, PPP1R14A, PPP1R14B, PPP2R1B, PPP2R2C, PRKAR2B, PRKCA, RAPGEF3, RASD1, RRAS2, TLN2
HIF1α Signaling	4.31	−4.62	ADM, AKT3, BMP6, CAMK1, EDN1, EGF, EGLN3, FGF2, FLT4, FOXP3, HGF, HSPA6, IGF1, IL6, IL6R, JUN, KDR, LDHB, MAP2K6, MET, MMP1, MMP10, MMP11, MMP13, MMP24, MMP28, MMP3, MMP9, MRAS, PIK3C2G, PIK3R1, PLCG2, PRKCA, PRKD1, PRKD3, RASD1, RRAS2, SLC2A4, TF, TGFA, VEGFD, VIM
Integration of Energy Metabolism	4.73	−3.30	ACSL4, ADCY4, ADCY5, ADIPOQ, ADRA2A, CACNB3, CD36, GNAI1, GNG11, GNG12, GNG2, GNG7, ITPR1, ITPR2, KCNB1, KCNJ11, MLXIPL, PFKFB1, PLCB1, PPP2R1B, PRKAR2B, PRKCA, RAPGEF3, STXBP1, VAMP2
AMPK Signaling	3.17	−3.15	ACACB, ADIPOQ, ADRA1A, ADRA2A, ADRB1, ADRB2, AK4, AK5, AK8, AKT3, CCNA2, CREB3L1, CREB3L4, CREB5, FOXO1, FOXO4, GNAI1, GNAL, GNAZ, GNG11, GNG12, GNG2, GNG7, IRS1, IRS2, KAT2B, LEP, LIPE, MRAS, PFKFB1, PFKFB3, PIK3C2G, PIK3R1, PPARGC1A, PPM1F, PPM1J, PPM1L, PPP2R1B, PPP2R2C, PRKAR2B, RAB3A, RAB9B, SLC2A4
RAR Activation	3.14	−4.56	ACVR1C, ACVR2A, ADCY4, ADCY5, ADH1B, ADH1C, AKR1C3, AKT3, ALDH1A1, ALDH1A2, ALDH1A3, CNGA1, COL10A1, COL1A1, COL1A2, COL3A1, CRABP2, CREB3L1, CREB3L4, CREB5, DHRS3, DRD2, DUSP1, EDA, EGF, FABP5, FOS, GUCY1A1, HOXA3, HOXA5, HOXD3, HOXD4, HSD17B6, IL17B, IL17D, IL33, IL6, JUN, KAT2B, KIT, KLF2, LEP, LIF, LIPE, LRAT, LTB, MAPK10, MEIS1, MEIS2, MMP1, MMP11, MMP13, MMP3, MMP9, MPPED2, NR2F1, NR2F6, NRIP2, OSM, PDE11A, PDE1A, PDE1B, PDE1C, PDE2A, PDE3A, PDE3B, PDE5A, PDE7B, PDE9A, PIK3C2G, PIK3R1, PPARG, PPARGC1A, PRKAR2B, RARB, RBP4, RBP7, RDH16, RDH5, RET, RHOJ, RHOQ, RHOU, RND1, RND3, RPS6KA2, RXRG, SDR16C5, SOCS3, STAT5A, STAT5B, STRA6, TGFBR2, TGFBR3, TNFSF12, TNFSF4, ZBTB16
Production of Nitric Oxide and Reactive Oxygen Species in Macrophages	3.23	−3.88	AKT3, APOC1, APOD, CAT, CLU, FOS, HOXA10, IFNGR1, JUN, MAP3K5, MAP3K8, MAPK10, NGFR, PIK3C2G, PIK3R1, PLCG2, PPARA, PPM1J, PPM1L, PPP1R14A, PPP1R14B, PPP2R1B, PPP2R2C, PRKCA, PRKD1, PRKD3, RBP4, RHOJ, RHOQ, RHOU, RND1, RND3, S100A8, SIRPA, TLR4, TNFRSF1B
Glutaminergic Receptor Signaling Pathway	3.32	−5.25	ADCY4, ADCY5, AKT3, CACNA2D1, CACNB3, CACNG4, CREB3L1, CREB3L4, CREB5, GABRB3, GABRD, GABRE, GABRP, GNAI1, GNAL, GPLD1, GRIA4, GRIK5, GUCY1A1, ITPR1, ITPR2, LCAT, NR3C1, PIK3C2G, PIK3R1, PLA2G4A, PLA2G5, PLA2R1, PLAAT3, PLAAT5, PLB1, PLCB1, PLCE1, PLCG2, PLCH2, PLCL2, PLD1, PNPLA2, PRKAR2B, PRKCA, PRKD1, PRKD3, SCN2A, SCN2B, SCN3A, SCN3B, SCN4A, SCN4B, SCN7A, SLC1A3, SLC1A7, SLC38A5, STX1B, TRPC1, VAMP2
Glycosaminoglycan Metabolism	3.16	−2.06	B3GNT3, B4GALT6, BGN, CHPF, CSGALNACT1, DCN, DSEL, FMOD, GPC3, HMMR, HPSE2, LYVE1, OGN, OMD, PRELP, SDC1, ST3GAL6, UST, VCAN
p53 Signaling	2.45	0.24	AKT3, BBC3, BIRC5, CCND2, CDKN2A, CHEK1, E2F1, GADD45G, JUN, KAT2B, PCNA, PIK3C2G, PIK3R1, PLAGL1, PMAIP1, SERPINB5, SNAI2, TP53AIP1, TP63, TRIM29
PPAR Signaling	2.33	2.52	FOS, IL1R1, IL1RL2, IL33, JUN, MRAS, NGFR, NR2F1, PDGFA, PDGFD, PDGFRA, PPARA, PPARG, PPARGC1A, PTGS2, RASD1, RRAS2, STAT5A, STAT5B, TNFRSF1B, TRAF2
Autophagy	2.01	−1.50	AKT3, BMP6, CREB3L1, CREB3L4, CREB5, DAPK2, E2F1, EGF, FGF2, FOS, FOXO1, GABARAPL1, HGF, IGF1, IRS1, IRS2, JUN, MAP1LC3C, MAPK10, MYC, NGFR, NOD2, PIK3C2G, PIK3R1, PPM1J, PPM1L, PPP2R1B, PPP2R2C, PRKAR2B, RAB7B, SESN1, SLC7A5, TGFA, TLR4, TNFRSF1B
PI3K/AKT Signaling	1.62	−0.90	AKT3, FOXO1, GDF15, GHR, IFNLR1, IL11RA, IL17RD, IL1R1, IL1RL2, IL20RA, IL22RA1, IL6R, ITGA1, ITGA10, ITGA6, ITGA7, ITGA9, ITGB3, ITGB4, ITGB8, MAP3K5, MAP3K8, MRAS, PIK3R1, PPM1J, PPM1L, PPP2R1B, PPP2R2C, PTGS2, RASD1, RRAS2
Triacylglycerol Degradation	0.77	−2.64	AADAC, ABHD6, ALDH2, CES1, LIPE, LPL, MGLL, PLB1, PNPLA2
Triglyceride Metabolism	1.18	−2.64	CAV1, FABP4, FABP5, LIPE, LPIN1, MGLL, PLIN1
mTOR Signaling	0.54	−3.71	AKT3, EIF3L, EIF4A1, GPLD1, IRS1, MRAS, PIK3C2G, PIK3R1, PLD1, PPM1J, PPM1L, PPP2R1B, PPP2R2C, PRKCA, PRKD1, PRKD3, RASD1, RHOJ, RHOQ, RHOU, RND1, RND3, RPS6KA2, RPS6KA3, RRAS2, VEGFD
Xenobiotic Metabolism AHR Signaling Pathway	0.33	−2.53	ABCG2, ALDH1A1, ALDH1A2, ALDH1A3, ALDH1L1, ALDH2, GSTM2, GSTP1, HDAC4, IL6
Sirtuin Signaling Pathway	0.32	−1.17	ABCA1, ACADL, ACSS2, CPS1, DUSP6, E2F1, EPAS1, FOXO1, FOXO4, GABARAPL1, GADD45G, IDH2, JUN, LDHB, LDHD, MAP1LC3C, MAPK15, MYC, PCK1, PFKFB3, PPARA, PPARG, PPARGC1A, RARB, SOD2, SOD3, TUBA1C, TUBA3C/TUBA3D
Mitochondrial Biogenesis	0.41	−2.12	ACSS2, CHD9, IDH2, MEF2C, PPARA, PPARGC1A, PPARGC1B, SOD2
Mitochondrial Dysfunction	0.46	−1.49	ACADL, ATP1A2, ATP1B2, BBC3, CACNA2D1, CACNB3, CACNG4, CAPN11, CAPN9, CLIC2, COX6C, COX7A1, CREB3L1, CREB3L4, CREB5, FBXW7, GPX3, GSTM2, GSTP1, HAP1, IDH2, ITPR1, ITPR2, LRRK2, MAOA, MAOB, MAP3K5, MAPK10, PIK3C2G, PIK3R1, PPARG, PPARGC1A, PRKAR2B, PRKN, RAPGEF3, SNCA, SOD2
TCA Cycle and Respiratory Electron Transport	0.62	−1.41	ADHFE1, IDH2, LDHB, ME1, ME3, PDK4, PDP2, SLC16A3

**Table 2 cells-13-01474-t002:** Energy-sensing molecules and their sensors.

Energy-Sensing Molecules	Molecules Whose Changes in Concentration Are Sensed
AMP-activated protein kinase (AMPK)	AMP and glycogen
Cytosolic guanylyl cyclase (cGC) and Basic helix loop helix-leucine zipper partner for the Max-like protein, Mlx (MondoA)	ATP
Guanylyl cyclase	cGMP
Phosphatidylinositol-5-phosphate 4-kinase-β (PI5PK4β)	GTP
Hypoxia-inducible factor 1 (HIF1)	Molecular oxygen (O_2_)
Peroxisome proliferator-activated receptors (PPARS)	Intracellular free fatty acids
Sirtuin 1, 3 (Sirt1 and Sirt3) proteins and AMPK	NAD^+^
Glucokinase	Glucose

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
