# Peer review of "Unraveling the Mystery of Energy-Sensing Enzymes and Signaling Pathways in Tumorigenesis and Their Potential as Therapeutic Targets for Cancer"

_cells, 2024, doi:10.3390/cells13171474_

Round 1

Reviewer 1 Report

Comments and Suggestions for Authors

The reviewer appreciates the opportunity to review the manuscript entitled "Unraveling the mystery of energy-sensing enzymes and signalling pathways in tumorigenesis and their potential as therapeutic targets for cancer" authored by Zeenat Mirza and Sajjad Karim. In the present manuscript, the authors reviewed the literature on biomolecules and pathways that are related to cancer with special emphasis on their energy-homeostasis significance.

Overall, the review offers a good and fast-reading reference for the research community involved in cancer studies. However, several points must be addressed before this manuscript achieves the publication's quality. Here, I offer recommendations to enhance the overall quality of the work:

* Please, define properly all the abbreviations that you use. For instance: “log2FC”

* Is it possible to separate the reviewed material for proteins/genes/pathways into different subsections? Currently, all this content looks entangled.

* Figures have poor quality. I understand that these diagrams should include many labels but at least the figures must have high-resolution. Please, increase the resolution to the standard values for publication. Also, in the labels of figures, for instance Figure 1, pathways are mentioned such as AKT -> NFkB … it would be instructive if, for the path, you use different line/font sizes to easily visualize the relevant elements in the pathway.

* In the context of line 202: “kinase is an enzyme” I would suggest “a kinase is an enzyme” because kinase is a family of enzymes. 

*  The title of the present review article “Unraveling the mystery of energy-sensing enzymes …” suggests to me that the mechanism for sensing energy changes by enzymes will be explained. However, from my reading that mechanism is not clarified for the proteins mentioned. For instance, how does AdK “sense” the energetic changes in the surroundings? As far as I understand, AdK only transfers a phosphate group and doesn’t “sense” the environment.  This needs to be clarified for the considered proteins in this review, and if several mechanisms are used propose general mechanisms.

* For a review article, I would consider it more useful to have a broader spectrum of references which looks limited in the present manuscript.

General comments:

* Please review the English grammar for the whole manuscript. Just as an example: “explore possibility” on line 24.

* Review all cited literature. As an example, in reference 2 one author is missing. 

* On line 386: is “v600EBRAF” using the correct fonts?

 * On line 98: “pathway etc” please write all the pathways explicitly.

* On line 50: “Metabolic and energetic … but not explored much” can be rephrased more professionally. 

Comments on the Quality of English Language

Extensive English grammar modifications are needed.

Author Response

REVIEWER 1

Comments and Suggestions for Authors

The reviewer appreciates the opportunity to review the manuscript entitled "Unraveling the mystery of energy-sensing enzymes and signalling pathways in tumorigenesis and their potential as therapeutic targets for cancer" authored by Zeenat Mirza and Sajjad Karim. In the present manuscript, the authors reviewed the literature on biomolecules and pathways that are related to cancer with special emphasis on their energy-homeostasis significance.

Overall, the review offers a good and fast-reading reference for the research community involved in cancer studies. However, several points must be addressed before this manuscript achieves the publication's quality. Here, I offer recommendations to enhance the overall quality of the work:

Response: Thank you for critical review and suggestions. We carefully addressed the comments below.

Comment 1: Please, define properly all the abbreviations that you use. For instance: “log2FC”

Response: Appreciate the comment. log2FC stands for log2 fold change, abbreviation has been defined at its first instance on page 89.

Comment 2: Is it possible to separate the reviewed material for proteins/genes/pathways into different subsections? Currently, all this content looks entangled.

Response: Agree but seems hard to separate as different subsections for proteins/genes/pathways for each case, so overlaps are unavoidable. A pathway involves the interactions between chemicals, proteins, and genes which integrate seemingly and helps to holistically understand their significance in a particular context. We tried our very best to separate as much possible.

Section 4 discusses mainly the genes/proteins associated with energy-sensing

Section 5 deals with the associated pathways

Comment 3:  Figures have poor quality. I understand that these diagrams should include many labels but at least the figures must have high-resolution. Please, increase the resolution to the standard values for publication. Also, in the labels of figures, for instance Figure 1, pathways are mentioned such as AKT -> NFkB … it would be instructive if, for the path, you use different line/font sizes to easily visualize the relevant elements in the pathway.

Response: Figures at high resolution (at least 300 dpi) have been generated using IPA and looks complicated as the number of participating protein molecules in a particular pathway are high and hence, appear intertwined.

Comment 4: In the context of line 202: “kinase is an enzyme” I would suggest “a kinase is an enzyme” because kinase is a family of enzymes.

Response: Correction done, to “kinases are enzymes”

Comment 5:   The title of the present review article “Unraveling the mystery of energy-sensing enzymes …” suggests to me that the mechanism for sensing energy changes by enzymes will be explained. However, from my reading that mechanism is not clarified for the proteins mentioned. For instance, how does AdK “sense” the energetic changes in the surroundings? As far as I understand, AdK only transfers a phosphate group and doesn’t “sense” the environment.  This needs to be clarified for the considered proteins in this review, and if several mechanisms are used propose general mechanisms.

Response: Thanks for the suggestion, we clarified as suggested (see section 4.1)

Adenosine kinase (ADK), an evolutionarily conserved phosphotransferase converts the purine ribonucleoside adenosine into 5′-adenosine-monophosphate (AMP). ADK incessantly monitors ATP, ADP, AMP - and other adenyl phosphate nucleotide levels inside the cell thereby regulating the amount of energy expended at the cellular level and hence, plays an important role in cellular energy homeostasis (Boison et al 2013, 2021).

Comment 6:  For a review article, I would consider it more useful to have a broader spectrum of references which looks limited in the present manuscript.

Response: Agreed, several new references have been added as suggested. The total number of references added now counts to 17. Following are the new additions:

Hernández Borrero LJ, El-Deiry WS. Tumor suppressor p53: Biology, signaling pathways, and therapeutic targeting. Biochim Biophys Acta Rev Cancer. 2021 Aug;1876(1):188556.

Shen J, Wang Q, Mao Y, Gao W, Duan S. Targeting the p53 signaling pathway in cancers: Molecular mechanisms and clinical studies. MedComm. 2023 May 28;4(3):e288.

Gonzalez-Flores D, Gripo A-A, Rodríguez A-B, Franco L. Consequences of Glucose Enriched Diet on Oncologic Patients. Applied Sciences. 2023; 13(5):2757.

Ferreira T, Rodriguez S. Mitochondrial DNA: Inherent Complexities Relevant to Genetic Analyses. Genes. 2024; 15(5):617.

Rodriguez OC, Choudhury S, Kolukula V, Vietsch EE, Catania J, Preet A, Reynoso K, Bargonetti J, Wellstein A, Albanese C, Avantaggiati ML. Dietary downregulation of mutant p53 levels via glucose restriction: mechanisms and implications for tumor therapy. Cell Cycle. 2012 Dec 1;11(23):4436-46.

Boison D. Adenosine kinase: exploitation for therapeutic gain. Pharmacol Rev. 2013 Apr 16;65(3):906-43. doi: 10.1124/pr.112.006361. PMID: 23592612; PMCID: PMC3698936.

Boison D, Jarvis MF. Adenosine kinase: A key regulator of purinergic physiology. Biochem Pharmacol. 2021 May;187:114321. doi: 10.1016/j.bcp.2020.114321. Epub 2020 Nov 6. PMID: 33161022; PMCID: PMC8096637.

Choi YK, Park KG. Targeting Glutamine Metabolism for Cancer Treatment. Biomol Ther (Seoul). 2018 Jan 1;26(1):19-28. doi: 10.4062/biomolther.2017.178. PMID: 29212303; PMCID: PMC5746034.

Yang L, Venneti S, Nagrath D. Glutaminolysis: A Hallmark of Cancer Metabolism. Annu Rev Biomed Eng. 2017;19:163–194. doi: 10.1146/annurev-bioeng-071516-044546.

Jin, J., Byun, JK., Choi, YK. et al. Targeting glutamine metabolism as a therapeutic strategy for cancer. Exp Mol Med 55, 706–715 (2023). https://doi.org/10.1038/s12276-023-00971-9

Wang Y, Luo J, Alu A, Han X, Wei Y, Wei X. cGAS-STING pathway in cancer biotherapy. Mol Cancer. 2020 Sep 4;19(1):136. doi: 10.1186/s12943-020-01247-w.

Motwani M, Pesiridis S, Fitzgerald KA. DNA sensing by the cGAS-STING pathway in health and disease. Nat Rev Genet. 2019;20:657–674.

Iurescia S, Fioretti D and Rinaldi M (2018) Targeting Cytosolic Nucleic Acid-Sensing Pathways for Cancer Immunotherapies. Front. Immunol. 9:711. doi: 10.3389/fimmu.2018.00711

Shi X., Young S., Cai K., Yang J., and Morahan G. (2022). Cancer susceptibility genes: update and systematic perspectives. The Innovation 3(5), 100277. https://doi.org/10.1016/j.xinn.2022.100277

Huang S., He C., Li J., et al., (2023). Emerging paradigms in exploring the interactions among diet, probiotics, and cancer immunotherapeutic response. The Innovation 4(4), 100456. https://doi.org/10.1016/j.xinn.2023.100456

Ren W., Ban J., Xia Y., et al., (2023). Echinacea purpurea-derived homogeneous polysaccharide exerts anti-tumor efficacy via facilitating M1 macrophage polarization. The Innovation 4(2), 100391. https://doi.org/10.1016/j.xinn.2023.100391

Zhang Z., Gao Q., Ren X., et al., (2023). Characterization of intratumor microbiome in cancer immunotherapy. The Innovation 4(5), 100482. https://doi.org/10.1016/j.xinn.2023.100482

Comment 7:  Please review the English grammar for the whole manuscript. Just as an example: “explore possibility” on line 24.

Response: Corrected as- explore “the” possibility.

English grammar and language have been thoroughly reviewed and improved.

Comment 8:  Review all cited literature. As an example, in reference 2 one author is missing.

Response: Referencing has been done using Clarivate’s EndNote and all the references have been reviewed as suggested. Thanks for the comment, however, we cross-checked; all the authors are there in reference 2 citation.

PubMed citation: Bray F, Ferlay J, Soerjomataram I, Siegel RL, Torre LA, Jemal A.

Present review article: Bray, F., J. Ferlay, I. Soerjomataram, R. L. Siegel, L. A. Torre and A. Jemal.

Comment 9:  On line 386: is “v600EBRAF” using the correct fonts?

Response: Corrected, the name V600E describes the nature and location of the mutation: V and E represent the specific amino acid that is mutated. In this case, valine (V) got replaced by glutamic acid (E). 600 refers to the location of the mutation: amino acid number 600 inside the protein. BRAF is italicized as it’s a gene name.

Comment 10:  On line 98: “pathway etc” please write all the pathways explicitly.

Response: Pathways written as advised:

Numerous signalling pathways like PI3K/AKT signaling, LKB1/AMPK signaling, RTK signaling, fatty acid oxidation pathway, glucose metabolism, glutamine metabolism, folate cycle, branched chain amino acid metabolism, etc. further contribute to the Warburg Effect and other metabolic phenotypes of cancer cells (Schiliro and Firestein 2021).

Comment 11:  On line 50: “Metabolic and energetic … but not explored much” can be rephrased more professionally.

Response: Rephrased as advised.

Energy linked metabolic changes in cancer cells were noticed much before the discovery of oncogenes and tumor suppressors, however, not explored much.

Comments on the Quality of English Language

Extensive English grammar modifications are needed.

Response: Thorough editing and language improvement has been done.

Reviewer 2 Report

Comments and Suggestions for Authors

This article provides a systematic review of common energy mediators, energy sensors, their metabolic properties, mechanisms, and signaling pathways involved in carcinogenesis, and explores the potential for identifying drugs that inhibit the energy metabolism of tumor cells. However, for the article to be accepted, please address and answer the following questions:

1. Please review the language and expression throughout the text, and ensure that the font size for each title is consistent (e.g., 6.2, etc.).

2. In page 14, paragraph 5.5, the content description is insufficient. Please expand on this section appropriately. For reference, see The Innovation (2022), 3(5), 100277 and The Innovation (2023), 4(4), 100456.

3. Most of the literature cited in this paper is outdated. Please try to reference literature from the past five years where possible. For reference, see The Innovation (2023), 4(2), 100391, The Innovation (2023), 4(4), 100452, and The Innovation (2023), 4(5), 100482, among other relevant sources.

4. In section 6.3, nucleic acid sensing is a critical mechanism in tumor immunotherapy. Please provide at least five examples to illustrate this point.

Comments on the Quality of English Language

 English is OK.

Author Response

REVIEWER 2

Comments and Suggestions for Authors

This article provides a systematic review of common energy mediators, energy sensors, their metabolic properties, mechanisms, and signaling pathways involved in carcinogenesis, and explores the potential for identifying drugs that inhibit the energy metabolism of tumor cells. However, for the article to be accepted, please address and answer the following questions:

Comment 1: Please review the language and expression throughout the text and ensure that the font size for each title is consistent (e.g., 6.2, etc.).

Response: We thoroughly reviewed and edited wherever possible. Font size has been made consistent.

Comment 2:  In page 14, paragraph 5.5, the content description is insufficient. Please expand on this section appropriately. For reference, see The Innovation (2022), 3(5), 100277 and The Innovation (2023), 4(4), 100456.

Response: We checked the suggested references. The reviewer has suggested us to include content from following articles: The Innovation (2022), 3(5), 100277 and The Innovation (2023), 4(4), 100456. We added both the suggested articles as citations:

 Shi X., Young S., Cai K., Yang J., and Morahan G. (2022). Cancer susceptibility genes: update and systematic perspectives. The Innovation 3(5), 100277. https://doi.org/10.1016/j.xinn.2022.100277

Huang S., He C., Li J., et al., (2023). Emerging paradigms in exploring the interactions among diet, probiotics, and cancer immunotherapeutic response. The Innovation 4(4), 100456. https://doi.org/10.1016/j.xinn.2023.100456

Biomarker discovery was facilitated by the breakthrough of widespread advances in whole-exome, whole-genome and customized panel of cancer-associated target genes sequencing allowing genotyping, Population-based genome-wide association studies and identification of single nucleotide variants (Shi et al 2022).  

Gut microbiota might be playing significant roles in cancer development and tumor microenvironment. Studying gut microbial metagenome and intratumor microbiome has opened up new avenues to understand the interactions among diet, probiotics, cancer and immunotherapeutic response and seems an emerging research paradigm for future exploration for anti-cancer immunotherapies (Huang et al 2023; Zhang et al 2023).

Comment 3: Most of the literature cited in this paper is outdated. Please try to reference literature from the past five years where possible. For reference, see The Innovation (2023), 4(2), 100391, The Innovation (2023), 4(4), 100452, and The Innovation (2023), 4(5), 100482, among other relevant sources.

Response: All the cited references have been checked for their significance and almost more than a dozen new citations including two from the suggested ones have been added.

Ren W., Ban J., Xia Y., et al., (2023). Echinacea purpurea-derived homogeneous polysaccharide exerts anti-tumor efficacy via facilitating M1 macrophage polarization. The Innovation 4(2), 100391. https://doi.org/10.1016/j.xinn.2023.100391

Zhang Z., Gao Q., Ren X., et al., (2023). Characterization of intratumor microbiome in cancer immunotherapy. The Innovation 4(5), 100482. https://doi.org/10.1016/j.xinn.2023.100482

Gut microbiota might be playing significant roles in cancer development and tumor microenvironment. Studying gut microbial metagenome and intratumor microbiome has opened up new avenues to understand the interactions among diet, probiotics, cancer and immunotherapeutic response and seems an emerging research paradigm for future exploration for anti-cancer immunotherapies (Huang et al 2023; Zhang et al 2023).

Macrophage modulation through transcriptional rewiring using phytocompounds that affects glycolysis and OXPHOS of M1 macrophages to boost IL-1b production has been shown effective for macrophage-associated inflammatory diseases like cancer (Ren et al 2023).

However, we found the following irrelevant to the present theme; hence, couldn’t add to the manuscript.

Kong Y., Yu J., Ge S., et al., (2023). Novel insight into RNA modifications in tumor immunity: Promising targets to prevent tumor immune escape. The Innovation 4(4), 100452. https://doi.org/10.1016/j.xinn.2023.100452

Comment 4: In section 6.3, nucleic acid sensing is a critical mechanism in tumor immunotherapy. Please provide at least five examples to illustrate this point.

Response: The section has been expanded as suggested with some new citations too.

6.3. Nucleic acid sensing

Nucleic acid sensing is a crucial mechanism in gene and immunotherapies that target cancer through therapeutic nucleic acids or genetically engineered immune cells. Nucleic acid sensing exerts both pro- and antitumor effects at different stages of tumorigenesis, supports immune cells in priming desirable immune responses during therapies and impacts the competence of gene therapy by inhibiting translation (Kong, Kim et al. 2023). Cyclic guanosine mono-phosphate-adenosine monophosphate synthase (cGAS) senses accumulated DNA in cells and gets activated upon binding to dsDNA. In an energy-consuming process, cGAS transforms ATP and GTP into cyclic GMP–AMP (cGAMP), a secondary messenger. cGAMP with other cyclic dinucleotides convey the signal downstream to the endoplasmic reticulum protein named stimulator of interferon genes (STING). Then, the signal cascades and finishes in interferon regulatory factor 3 (IRF3) and NF-κB targets in the nucleus, leading to the secretion of type-I interferon which is vital to tumor-specific T cells (Motwani 2019; Wang Y 2020). Similarly, the RIG-I-like receptors (RLRs)-MAVS pathways, responsible for cytosolic RNA sensing have been targeted for cancer treatment in preclinical translational research (Iurescia 2018). Comprehending the interplay of distinct nucleic acid sensing mechanisms can facilitate the creation of novel strategies for optimization of cancer immunotherapy.

Comments on the Quality of English Language

English is OK.

Response: Thank you.

Round 2

Reviewer 1 Report

Comments and Suggestions for Authors

In the present version of the manuscript "Unraveling the mystery of energy-sensing enzymes and signalling pathways in tumorigenesis and their potential as therapeutic targets for cancer" the authors Zeenat Mirza and Sajjad Karim considerably improved its quality and readability, for instance by doing a more extensive literature search on the topic and reviewing the English grammar of the whole manuscript. However, I still have a major concern in considering this work for publication which is the quality of Figures 3 and 4: one cannot read the labels of these Figures and therefore they are unuseful for readers. Thus, the quality of these figures must be improved, one possibility could be that instead of showing the complete picture in a single figure, the authors divide it into four parts and show the parts in separate Figures 3a, 3b, .. (same for Figure 4). This could help keep the initial figure's quality (thrown most likely by existing software) in each part. 

Comments on the Quality of English Language

Minor English editing is needed.

Author Response

Comments: In the present version of the manuscript "Unraveling the mystery of energy-sensing enzymes and signalling pathways in tumorigenesis and their potential as therapeutic targets for cancer" the authors Zeenat Mirza and Sajjad Karim considerably improved its quality and readability, for instance by doing a more extensive literature search on the topic and reviewing the English grammar of the whole manuscript. However, I still have a major concern in considering this work for publication which is the quality of Figures 3 and 4: one cannot read the labels of these Figures and therefore they are unuseful for readers. Thus, the quality of these figures must be improved, one possibility could be that instead of showing the complete picture in a single figure, the authors divide it into four parts and show the parts in separate Figures 3a, 3b, .. (same for Figure 4). This could help keep the initial figure's quality (thrown most likely by existing software) in each part. 

Response: We appreciate your kind words and good judgment skills. We have replaced both figure 3 and 4 having better appearance as suggested.

Round 3

Reviewer 1 Report

Comments and Suggestions for Authors

In the present version of the manuscript "Unraveling the mystery of energy-sensing enzymes and signalling pathways in tumorigenesis and their potential as therapeutic targets for cancer" the authors Zeenat Mirza and Sajjad Karim have improved the quality of the manuscript, especially in the Figures. I have no further comments on this manuscript and therefore accept it for publication in its present form.

Comments on the Quality of English Language

English grammar would need minor editing.